# MIXTURE OF COMPLEMENTARY AGENTS FOR ROBUST LLM ENSEMBLE

## ABSTRACT

Multi-AI collaboration—such as ensembling or debating large language models (LLMs)—is a promising paradigm for aggregating information and boosting performance. A foundational step in these pipelines is to feed the responses of several *proposer* LLMs into a *summarizer* LLM, which synthesizes a better answer. However, choosing which proposers to include is non-trivial. Existing approaches primarily focus either on accuracy (picking the strongest models) or diversity (ensuring variety), and often overlook the interactions among proposers and with the summarizer. We introduce *complementary-MoA*, a principled framework for proposer selection built on the notion of complementarity: the value of a proposer lies not only in its individual performance, but in how it improves the joint performance of the ensemble. Leveraging a small training set with ground truth answers, we propose several greedy-based algorithms that explicitly optimize for complementarity while offering accuracy–efficiency trade-offs for proposer selection. Empirically, we demonstrate why accuracy- and diversity-seeking heuristics are fundamentally flawed in LLM ensembles, and validate the robustness and superiority of our complementarity-based methods.

## 1 INTRODUCTION

As today's Large language model (LLM) ecosystem fragments into numerous models with diverse expertise, collaboration among LLMs has become promising and sometimes necessary for tackling emerging tasks such as mathematical reasoning (Du et al., 2023), code generation (Mahmud et al., 2025), and complex decision-making (Wu et al., 2023). A convenient instantiation is *ensemble after inference*, which aggregates the LLM outputs after the generation of full responses. This includes well-studied frameworks such as *LLM debating* (Du et al., 2023; Estornell & Liu, 2024; Chan et al., 2023), in which multiple models iteratively exchange arguments before a final decision is reached, and *mixture-of-agents (MoA)* (Wang et al., 2024; Li et al., 2025), which uses layered and summarization schemes to combine diverse model outputs.

A fundamental step in the ensemble framework is inputting $N$ LLM responses—the *proposers*—into an aggregating LLM—the *summarizer*—which synthesizes a potentially better answer. Selecting which proposers to include is therefore critical: for a large proposer pool, it is impractical and inefficient to input responses from every available model due to context-window limits and the degraded inference ability (Liu et al., 2023). Existing methods often choose a small set of proposers based on their independent performance, following two heuristics: (i) *accuracy-seeking*—prioritize high-accuracy proposers or even a single top model with multiple samples (Li et al., 2025; Jiang et al., 2023), and (ii) *diversity-seeking*—explicitly mix heterogeneous outputs or prompts to avoid reinforcing similar mistakes (Lau et al., 2024; Wang et al., 2024).

However, both heuristics overlook team effects, a key determinant of LLM ensemble performance. In particular, accuracy-seeking methods rank proposers only by their individual performance, while diversity-seeking methods reward variance regardless of quality. We instead propose *mixture-of-complementary-agents (**complementary-MoA**)*—a framework that selects proposers for how well they work together as a team and with the summarizer. The importance of complementarity can be observed from Fig. 1, which compares summarizer accuracy when inputting (i) the individually most accurate proposer versus (ii) the proposer that most complements the summarizer. In this example, we consistently observe a nontrivial gap between the two choices, and furthermore, the

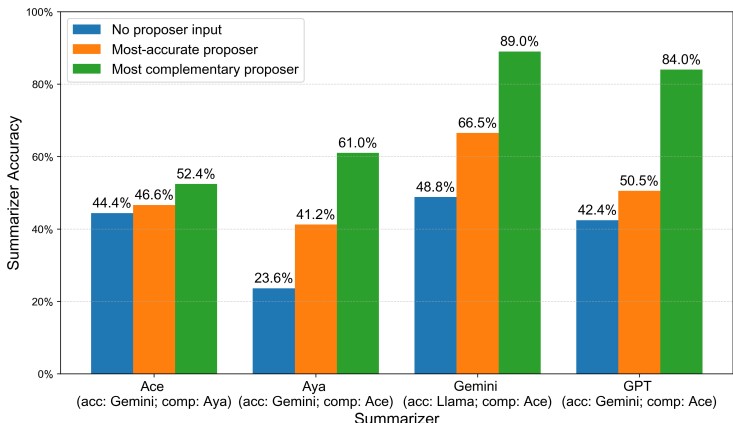

Figure 1: Summarizer accuracies on AIME (dolbokostya, 2025) when inputting the **most accurate** proposer vs. the **most complementary** proposer. For each summarizer $s$, the proposer pool is {Qwen3-32B, Sky-T1-32B-Preview, Aya-expanse-32B, Gemini-1.5-Pro, Llama-3.3-70B-Instruct, AceReason-Nemotron, and GPT-4o}, excluding $s$ itself.

most-complementary proposer is often weak on its own. The upshot is both promise and challenge: as the ensemble size $k$ (the number of proposers selected for input) grows, complementarity-based selection can yield substantial gains, yet it also complicates the search, since optimal teams cannot be inferred from individual performance alone.

In this paper, we seek efficient algorithms that select complementary proposers for better ensembling, using multiple-choice QA as an example. Exhaustively searching over all proposer combinations is often infeasible, especially considering the earlier observation that inputting responses from the same LLM under multiple prompts can boost performance (Li et al., 2025; Lau et al., 2024). To incorporate both intra-model and cross-model diversity, we pair each LLM with multiple instruction prompts and treat each model–prompt pair as a proposer. Formally, given a small labeled validation set, an LLM summarizer (oracle), and a target ensemble size $k$, our goal is to select $k$ out of $N$ proposers that maximize the ensemble accuracy while minimizing sample complexity, measured by summarizer calls.

Motivated by the analogy to feature selection with a black-box objective $\text{Acc}(S)$ (the ensemble accuracy given selected proposer set $S$), we develop proposer-selection algorithms that navigate the accuracy-efficiency trade-offs. First, we introduce a wrapper-style method that keeps the summarizer in the loop, called the *model-first greedy*. At each step, we first select the model whose prompt variants have the largest marginal contribution to $\text{Acc}(S)$ on average, and then add the best prompt instance from that model to $S$. Model-first greedy reduces sample complexity by prioritizing the selection of the most complementary model, rather than the model-prompt pair. To further reduce sample complexity, we propose two algorithms that consider label-level complementarity. *Truth-prediction greedy* selects proposers based on how well their reported labels help predict the ground truth; *oracle-surrogate greedy* first fits a simple surrogate of the oracle and then selects proposers based on their marginal contributions measured by the surrogate model. Both methods rely only on label-level statistics and therefore require no—or only light—summarizer calls.

We run extensive experiments across two task families (a multi-choice math dataset (dolbokostya, 2025) and a binary-choice causal-reasoning dataset (Jin et al., 2023)), spanning multiple proposer pools (a dominating-LLM regime and a mixed-crowd regime), various summarizers and ensemble sizes $k$. Our results confirm the fundamental limitations of the discussed baselines—for each baseline, we find settings where it performs poorly. In contrast, our complementarity-guided methods, including the truth-prediction greedy which requires no summarizer call, are consistently robust across all scenarios. Moreover, we frequently observe substantial gains from model-first greedy over the strongest baseline, underscoring that explicitly optimizing for complementarity is crucial in ensemble frameworks.

In summary, our main contributions are threefold:

- We identify complementarity as a key, yet overlooked objective in agent-level LLM ensembles and propose a more principled proposer-selection framework, called complementary-MoA, that explicitly optimizes it.

- We instantiate complementary-MoA for multiple-choice QA with three algorithms that realize different accuracy–efficiency trade-offs. Depending on task needs and computation budgets, these methods offer practical options for building more trustworthy and robust multi-AI collaboration systems.

- We validate the framework through extensive experiments on various datasets, proposer pools, summarizers, and ensemble sizes against a rich set of baselines. The results clarify when and why baseline heuristics succeed or fail, and demonstrate the robustness and effectiveness of our proposed methods across all tested settings.

## 2 PROBLEM STATEMENT

We consider a dataset of multiple-choice questions $\mathcal{Q}$, and each question $q \in \mathcal{Q}$ has a ground-truth label $Y_q \in \mathcal{Y}$. We assume true labels are available on a validation subset $\mathcal{Q}_T \subset \mathcal{Q}$ of size $m = |\mathcal{Q}_T|$, while the remaining questions require inference (test data).

There are $N$ *proposers*. Proposer $i$ provides for question $q$ a response $R_{i,q} = (X_{i,q}, Z_{i,q})$, where $X_{i,q} \in \mathcal{Y}$ is a proposed label and $Z_{i,q}$ is textual supporting reasoning (e.g., chain-of-thought reasoning). In our setting, we permit multiple proposers to originate from a single LLM by varying the prompt. This is inspired by prior studies (Li et al., 2025; Lau et al., 2024), showing that feeding multiple responses from the same model to the summarizer can benefit the ensemble. Let $n_{\mathrm{prompt}}$ and $n_{\mathrm{LLM}}$ be the number of prompts and models; the total number of proposers is then $N = n_{\mathrm{prompt}} \cdot n_{\mathrm{LLM}}$.

To improve accuracy, a *summarizer* aggregates multiple proposer responses, and outputs a potentially more accurate label. Due to practical constraints (e.g., LLMs often have strict input context limits), we aim to select a (small) subset of proposers for the ensemble. Formally, given the ensemble size $k$ and a subset $S \subseteq [N]$ with $|S| = k$, the summarizer outputs $f(\boldsymbol{R}_{S,q})$, where $\boldsymbol{R}_{S,q} = (R_{i,q})_{i \in S}$. Both proposer and summarizer outputs are stochastic, and the key design choice is which proposers to select as input to the summarizer.

We evaluate a selection $S$ by the summarizer accuracy test data:

$$\mathrm{Acc}_f(S) = \frac{1}{|\mathcal{Q} \backslash \mathcal{Q}_T|} \sum_{q \in \mathcal{Q} \backslash \mathcal{Q}_T} \Pr\left[ f(\boldsymbol{R}_{S,q}) = Y_q \right].$$

The central problem is to choose $k$ out of $N$ proposers to maximize accuracy, given summarizer $f$:

$$S^* = \arg \max_{S \subseteq [N], |S| = k} \mathrm{Acc}_f(S). \tag{1}$$

We study the trade-offs introduced by the choice of $k$, though for clarity, most of our analysis proceeds while supposing $k$ is fixed.

### 2.1 PREVIOUS IDEAS

**Label-only aggregation.** The simplest approach aggregates only the discrete answers and ignores textual rationales. A common choice is (weighted) majority voting over all proposers or a selected subset. When proposers are conditionally independent with known accuracies, decision theory implies that a weighted majority rule (with weights proportional to log-odds of correctness) is optimal (Nitzan & Paroush, 1982). Our setting departs from these assumptions: LLM proposers exhibit strong dependencies, and their rationales carry additional signal. Empirically, aggregation schemes that leverage an LLM summarizer to use rationales outperform simple majority vote on labels alone (Lau et al., 2024; Tekin et al., 2024). Our experiments further confirm this point (see Appendix B.1).

**Accuracy-seeking aggregation.** A widely used heuristic in LLM ensembling is to select proposers by their estimated individual accuracy, where the intuition is that proposers with higher accuracy contribute more reliable evidence on new instances. For example, one idea, called the *self-MoA*, is

to sample multiple diverse responses from the single best model and feed them to a summarizer (Li et al., 2025).

**Diversity-seeking aggregation.** A parallel line in the LLM ensemble literature argues that accuracy alone is insufficient: ensembles can benefit from diverse views. This intuition inspired several suggestions that explicitly encourage diversity or strike an accuracy–diversity trade-off (e.g., maximize diversity conditioned on an accuracy bar) (Lau et al., 2024; Tekin et al., 2024; Wang et al., 2024). However, for LLM ensembles, such diversity-first strategies can be counterproductive: by admitting weak proposers in the name of variety, they often introduce low-quality or correlated errors that depress the final aggregation performance.

## 3 METHODS

Our central idea is to select proposers based on their collaborative performance with each other and with the summarizer—the selected proposers should complement their teammates. In principle, one could exhaustively evaluate all size-$k$ teams and pick the subset that maximizes summarizer accuracy. In practice, searching over all $\binom{N}{k}$ subsets is typically infeasible—e.g., even with $N = 20$ and $k = 5$ there are 15,504 candidate teams—especially given the high inference cost of summarizing multi-rationale inputs.

An immediate idea is a **greedy algorithm**: we can iteratively find the proposer with the largest marginal contribution to the summarizer accuracy until we find $k$ proposers. In particular, we initialize $S_0 = \emptyset$, and for $t = 1, \ldots, k$, choose

$$i_t \in \arg \max_{i \in [N] \setminus S_{t-1}} \left[ \mathrm{Acc}(S_{t-1} \cup \{i\}) - \mathrm{Acc}(S_{t-1}) \right],$$

then update $S_t = S_{t-1} \cup \{i_t\}$.

The performance of the greedy algorithm depends on the submodularity of the accuracy function, which in turn depends on the summarizer. It turns out that for LLM summarizers, the accuracy function is not even monotone (and thus not submodelar)—including a low-accuracy proposer in the pool can actually reduce overall summarization performance. This observation is supported by prior work (Li et al., 2025) and our experiments in Appendix B.2. Therefore, in principle, the greedy algorithm can be far from the optimum in the worst case. However, as we will see, the empirical performance of the (simplified versions of) the greedy algorithm is generally robust and significantly outperforms the baselines.

A more detailed discussion of related work is deferred to Appendix A.

Although the greedy algorithm is conceptually simple, it can be computationally demanding: it requires evaluating the accuracy function $O(kN)$ times, which entails $O(kNm)$ calls to the summarizer. It is thus important to explore the trade-off between ensemble accuracy and efficiency via some heuristic variants. In our experiments, we implement only the simplified methods rather than the full greedy algorithm.

### 3.1 MODEL-FIRST GREEDY

Recall that a model and an instruction prompt determines a proposer. However, responses generated by different prompts of the same model are typically more similar than responses generated by different models under the same prompt (see Appendix B.1). Inspired by hierarchical feature selection (Ristoski & Paulheim, 2014), we introduce a simplification called *model-first greedy*. Unlike standard greedy—which estimates the marginal gain of every proposer using all $m$ questions at each iteration —model-first greedy scores all $n_{\mathrm{prompt}}$ proposers from the same model using the common set of $m$ questions, then chooses a proposer only within the best model. Concretely, in iteration $t$:

1. For each model $i \in [n_{\mathrm{LLM}}]$, estimate its average accuracy by randomly assigning each question in $\mathcal{Q}_T$ to one of its associated proposers and averaging over $m$ questions.

2. Select the model with the highest average accuracy, then pick the proposer associated with this model with the highest estimated accuracy in the previous step.

Intuitively, the procedure prioritizes model selection while allowing more randomness in proposer selection. This reduces summarizer calls per iteration from $N \cdot m$ to $n_{\text{LLM}} \cdot m$.

## 3.2 LABEL-LEVEL COMPLEMENTARITY

Model-first greedy estimates each proposer's marginal contribution via direct calls to the summarizer. However, the proposers' labels themselves carry predictive signals: the summarizer is more likely to answer correctly when it receives more correct inputs. This motivates the idea of selecting proposers based on their *label-level* information, which can improve scalability by avoiding extensive calls to the summarizer oracle. This idea is related to *filter-based* feature selection methods, e.g. (Peng et al., 2005; Urbanowicz et al., 2018), which remove likely weak features without retraining the predictor based on correlations between features. In particular, we use an alternative set function $\widehat{Acc}$, defined with respect to a label-based summarizer $g$, and use it to guide proposer selection. This yields the following two methods.

**Truth-Prediction Greedy** Built on the intuition that labels from a set of complementary proposers can predict the true label more accurately, we can train a light-weight machine learning model to predict $Y_q$, and use it to select informative proposers. Given a set of proposers $S$ and a family of models parametrized by $\theta \in \Theta$, we compute a value $\widehat{\text{Acc}}_{g_\theta}(S)$ using the following procedure:

1. Partition $\mathcal{Q}_T$ into a training set $\mathcal{Q}_T^{tr}$ and a validation set $\mathcal{Q}_T^{val}$ for cross validation.
2. **Fit $g_\theta$.** Use the data in the training set, $((X_{i,q})_{i \in S}, Y_q)_{q \in \mathcal{Q}_T^{tr}}$ to fit a model $g_\theta$ that maps $|S|$ labels on a question to a (hard) prediction of the ground truth label. Here, proposers' generated labels are viewed as features.
3. **Score proposer set $S$.** On the validation set, evaluate the accuracy of $g_\theta$ using responses from $S$ and return $\widehat{\text{Acc}}_{g_\theta}(S)$.

Next, we select proposers using a variant of the greedy algorithm, called the $k$-greedy (Alg. 1), using $\text{Acc}_{g_\theta}$ as the set function. We first initialize a set of proposers $S_0 = \emptyset$. Then, in round $t \in \{1, \ldots, k\}$, unlike standard greedy—which estimates a candidate's marginal gain relative to the current selected set $S_{t-1}$—$k$-greedy's estimation always conditions on a set of $k$ proposers. The intuition is that LLM summarizers are non-monotone, so an element that looks promising early can hurt performance at the final team size $k$. Concretely, given $S_{t-1}$, we randomly select $k - t + 1$ proposers, to form a team of size $k$, denoted as $L$. Then, we measure candidate $i$'s contribution ($i \notin S_{t-1}$) as the accuracy difference with and without $i$ (replacing one randomly chosen proposer in $L$). Averaging this difference over several random completions yields a more faithful estimate of $i$'s value at the final team size. We refer to this method as *truth-prediction greedy*, which applies the $k$-greedy algorithm to the set function $\widehat{\text{Acc}}g_\theta$. We emphasize that truth-prediction greedy relies on a lightweight ML model to guide proposer selection, but the final ensemble is still formed by feeding the chosen proposers into the summarizer.

**Oracle-Surrogate Greedy** Proposer selection under truth-prediction greedy does not depend on the summarizer, so it may diverge from the ensemble's true test performance. As an alternative approach, we propose *oracle-surrogate greedy*, where the idea is to fit a simple surrogate model to simulate the summarizer's behavior using a small number of oracle queries on the training set, then use the surrogate to score and select proposers. Although this method requires some summarizer calls for training, the surrogate model is kept simple as we only focus on label-level information, making it more sample-efficient than model-first greedy in practice.

We consider a surrogate model $\tilde{g}$ based on the assumption that the summarizer's accuracy depends primarily on how many of the $k$ input labels are correct. Specifically, $\tilde{g} : \{0, \ldots, k\} \to [0, 1]$ maps a count $c$ of correct labels to the expected summarizer accuracy when exactly $c$ out of $k$ inputs are correct. This implies that our surrogate model greatly reduces the sample complexity by not distinguishing the proposer ID. Given a set of proposers $S$, the following procedure returns a value $\widehat{\text{Acc}}_{\tilde{g}}(S)$ for set $S$:

1. Partition $\mathcal{Q}_T$ into a training set $\mathcal{Q}_T^{tr}$ and a validation set $\mathcal{Q}_T^{val}$ for cross validation.

---

**Algorithm 1:** $k$-Greedy Proposer Selection w.r.t. Acc

---

**Input:** ground set $[N]$, target size $k$, set function Acc, repetitions $M$
**Output:** selected set $S_k$
$S_0 = \emptyset$ ;                    // initialize the set of selected proposers
**for** $t = 1$ **to** $k$ **do**
    **for** $i \in [N] \setminus S_{t-1}$ **do**
        $\Delta_i = 0$;
        **for** $\tau = 1$ **to** $M$ **do**
            Sample $L \subseteq [N] \setminus (S_{t-1} \cup \{i\})$ uniformly with $|L| = k - |S_{t-1}|$;
            Pick $j \in L$ uniformly at random and set $L' \leftarrow (L \setminus \{j\}) \cup \{i\}$;
            $\Delta_i \mathrel{+}= \text{Acc}(S_{t-1} \cup L') - \text{Acc}(S_{t-1} \cup L)$;
        $\widehat{\Delta}_i(S_{t-1}) = \Delta_i/M$ ; // estimated marginal via random completions
    Choose $i^\star \in \arg\max_{i \in [N] \setminus S_{t-1}} \widehat{\Delta}_i(S_{t-1})$;
    $S_t = S_{t-1} \cup \{i^\star\}$;
**return** $S_k$;

---

2. **Fit $\tilde{g}$.** For each $c \in \{0, \ldots, k\}$, repeat $T_{\tilde{g}}$ times: (i) sample a question from $\mathcal{Q}_T^{\text{tr}}$ and a size-$k$ set of proposers whose responses contain exactly $c$ correct labels; (ii) query the summarizer on these $k$ responses. Define $\tilde{g}(c)$ as the empirical accuracy—i.e., the average correctness of the summarizer across the $T_{\tilde{g}}$ queries.

3. **Score proposer set $S$.** For each $q \in \mathcal{Q}_T^{val}$, compute $c_q(S)$, the number of correct labels in $S$, and assign $\widehat{\text{Acc}}_{\tilde{g}}(S) = \frac{1}{|\mathcal{Q}_T^{val}|} \sum_{q \in \mathcal{Q}_T^{val}} \tilde{g}(c_q(S))$.

Next, we select a set of $k$ agents by calling Alg. 1 with $\widehat{\text{Acc}}_{\tilde{g}}(S)$ as the set function.

## 4 EXPERIMENTS

In this section, we first introduce the experimental setups, then we validate the proposed complementary-MoA framework, diagnose the failure modes of baseline selectors, quantify efficiency–accuracy trade-offs, and finally study prompting strategies for the summarizer that yield stronger ensembles.

### 4.1 EXPERIMENT SETUPS

**Dataset**  We look for datasets with multi-choice reasoning questions. We choose two popular reasoning datasets: AIME (dolbokostya, 2025) and CLadder (Jin et al., 2023). AIME comprises about 1,600 curated mathematical problems and their answers sourced from prestigious competitions such as the American Invitational Mathematics Examination (AIME) and the International Mathematical Olympiad (IMO). The dataset was originally open-ended, with all true answers being integers. For each question, we randomly pick four integers between 0 and 1,000 to serve as additional incorrect answers, resulting in a five-choice QA dataset. CLadder contains 10k causal reasoning questions that translate queries from causal graphs into natural-language yes/no questions spanning association, intervention, and counterfactual levels. We sample 500 questions from AIME and 1k questions from CLadder for our experiments.

**Models**  We consider a diverse set of LLMs: QwQ-32B (Team, 2025b), Qwen3-32B (Team, 2025c), Sky-T1-32B-Preview (Team, 2025a), aya-expanse-32B (et al., 2024b), Gemini1.5-Pro (et al., 2024a), Llama-3.3-70B-Instruct (Dubey et al., 2024), AceReason-Nemotron (Chen et al., 2025a), GPT-4o (et al., 2024c). Each model is operated with a default temperature of 0.7. For each of the LLMs, we consider $n_{\text{prompt}} = 5$ different prompts which are presented in Appendix C, and each model-prompt pair is viewed as a proposer.

We evaluate ensemble performance across four factors—dataset, proposer pool, summarizer, and ensemble size $k$—using two datasets (CLadder, AIME), two pools (with/without QwQ), multiple

summarizers, and several choices of $k \leq 5$. Because QwQ typically has dominant performance, we use the "with QwQ" setting to simulate a dictator-style scenario, while the "without QwQ" reflects a mixed field. For example, "(AIME, with QwQ, Aya, $k = 3$)" denotes the AIME dataset, a pool including QwQ, Aya as summarizer, and selecting three proposers. We limit $k$ to 5 because larger ensembles show diminishing returns (Lau et al., 2024), while the cost of searching for the optimal team grows quickly.

## 4.2 A COMPARISON OF PROPOSER SELECTION METHODS

Based on previous ideas in Section 2.1, we consider the following baselines:

- **Input-all**: input all $N$ proposers.[1]
- **Best-model**: identify the most accurate model and select all proposers associated with it, in line with (Li et al., 2025).
- **Top-accuracy**: select the most accurate $k$ proposers overall.
- **MoA (per-model top-1)**: for each model, select the single most accurate proposer, inspired by the original mixture-of-agents framework (Wang et al., 2024).
- **Conditioned-diversity**: start with the most accurate proposer, then greedily add the proposer that maximizes average disagreement with the selected set, subject to an accuracy threshold $\tau$. This is inspired by (Lau et al., 2024).

We randomly select $m = 400$ questions for proposer selection and use the remaining questions for accuracy computing. For each LLM, we iteratively use $n_{\text{prompt}} = 5$ prompts to solicit responses for all the sampled questions, which returns $N = 40$ proposers' responses for each question. We randomize proposer order and include their individual accuracies in the instructions while inputting into the summarizer. For methods that require training, we feed the $m$ proposer-selection data into the selection algorithm, which returns a set of $k$ proposers that are evaluated on the test data. To further reduce the variance of the ensemble accuracy (due to the randomness caused by the default temperature of LLMs), we repeatedly call the summarizer ten times for each question and take the average.

Table 1 and 2 compare two settings with and without QwQ as proposers. The former reflects a "dictator" scenario in which QwQ dominates; the latter represents a mixed field with comparable proposers. For each method, we also report per-model selection counts, indicating the number of proposers selected from each model. We defer the results for other settings to the appendix.

**Importance of Complementarity** First, our results suggest that the accuracy-seeking or diversity-seeking baselines are not robust. On AIME, accuracy-seeking methods (Top-accuracy, Best-model) perform well when a single proposer is both the most accurate and the best collaborator with the summarizer (e.g., QwQ with Aya); their performance greatly drops when QwQ is removed from the proposer pool (Table 2). On the other hand, diversity-seeking methods (MoA, Conditioned-diversity) degrade when there is a dominant collaborator with the summarizer, as in Table 1.

In contrast, the label-level, complementarity-aware methods, Truth-prediction Greedy and Oracle-surrogate Greedy, while not always the top performers, deliver consistently strong performance across all settings. Furthermore, Model-first Greedy yields particularly large gains over all baselines—with almost 10% accuracy gain compared with the best baseline. These findings provide strong evidence that selecting proposers while explicitly considering complementarity is crucial for effective LLM ensembles.

*What explains this performance discrimination?* Figure 2 presents the empirical distributions of the number of correct labels $c \in \{0, \dots, k\}$ obtained by the selected proposers under three representative methods in the setting without QwQ as proposers. The overlaid curve shows the summarizer accuracy conditioned on $c$ correct labels.[2] Clear patterns emerge: accuracy-seeking baselines, such as Best-model, induce a U-shaped distribution of $c$, while diversity-seeking baselines, such as

---

[1]To fit the token limit, we truncate the responses from each proposer before summarizing.

[2]To reduce variance, we pool samples from all proposers to estimate the conditioned accuracy. Hence, the curve is identical across methods within the same setting.

Table 1: A comparison of methods in the (AIME, with QwQ, Aya, $k = 5$) setting.

| Method | Average counts of selected proposers | | | | | | | | Accuracy |
| --- | --- | --- | --- | --- | --- | --- | --- | --- | --- |
| | QwQ | Qwen | Llama | Gemini | GPT | Sky | Aya | Ace | |
| Input-all | 5 | 5 | 5 | 5 | 5 | 5 | 5 | 5 | 0.746 |
| Best-model | 5 | — | — | — | — | — | — | — | 0.742 |
| Top-accuracy | 3 | 1 | — | 1 | — | — | — | — | 0.729 |
| MoA | 1 | 1 | 1 | 1 | 1 | 1 | 1 | 1 | 0.635 |
| Conditioned-diversity | 1 | — | 1 | — | — | — | 1 | 2 | 0.546 |
| **Truth-prediction Greedy** | 2 | 2 | — | 1 | — | — | — | — | 0.715 |
| **Oracle-surrogate Greedy** | 2 | 1 | — | 1 | — | — | — | 1 | 0.702 |
| **Model-first Greedy** | 2 | 1 | — | — | — | — | — | 2 | **0.830** |

Table 2: A comparison of methods in the (AIME, without QwQ, Aya, $k = 5$) setting.

| Method | Average counts of selected proposers | | | | | | | Accuracy |
| --- | --- | --- | --- | --- | --- | --- | --- | --- |
| | Qwen | Llama | Gemini | GPT | Sky | Aya | Ace | |
| Input-all | 5 | 5 | 5 | 5 | 5 | 5 | 5 | 0.722 |
| Best-model | — | — | 5 | — | — | — | — | 0.369 |
| Top-accuracy | 1 | — | — | — | 1 | 2 | 1 | 0.614 |
| MoA (mixed) | 1 | 1 | 1 | 1 | 1 | 1 | 1 | 0.616 |
| Conditioned-diversity | 1 | 1 | — | — | 1 | 2 | — | 0.662 |
| **Truth-prediction Greedy** | 3 | — | 1 | 1 | — | — | — | 0.637 |
| **Oracle-surrogate Greedy** | 1 | 1 | 1 | — | 1 | — | 1 | 0.685 |
| **Model-first Greedy** | 3 | — | — | — | — | — | 2 | **0.815** |

Conditioned-diversity, exhibit a bell-shaped distribution. This indicates that Best-model tends to select proposers who make similar mistakes, which can be problematic when the summarizer accuracy curve is concave—i.e., when the marginal benefit of additional correct answers diminishes. However, Conditioned-diversity concentrates mass around $c = \lfloor k/2 \rfloor$ by seeking different proposers, which can be suboptimal when the summarizer requires a strong majority to achieve a significant accuracy boost. In contrast, complementarity-based methods yield distributions that lie between these two extremes, illustrating their robustness across different summarizer behaviors. Analogous figures for other methods are deferred to the appendix.

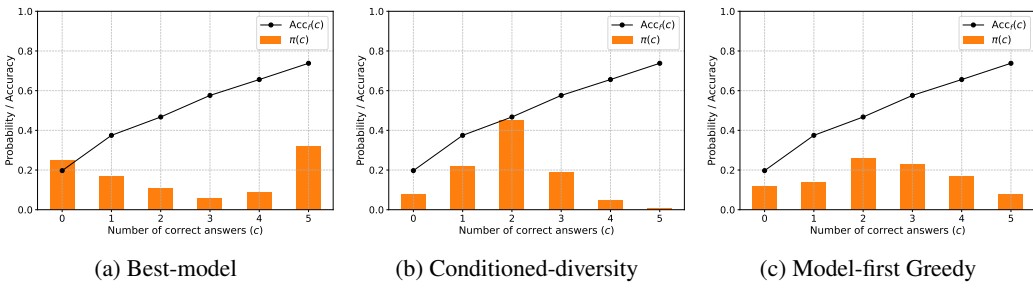

(a) Best-model  (b) Conditioned-diversity  (c) Model-first Greedy

Figure 2: Distribution of the number of correct answers (bars) and summarizer accuracy $\text{Acc}_f(c)$ (line) for three exemplary methods in the (AIME, without QwQ, Aya, $k = 5$) setting.

We report additional results on the binary-answer dataset CLadder in the appendix. Across methods, performance is largely similar—there are about 20% questions that no aggregation method is able to answer—with Model-free Greedy yielding only a marginal improvement. This reflects a regime where ensembling offers limited upside and underscores the need for robustness: while simple baselines may sometimes perform well, we demand methods that perform reliably across all settings.

**Efficiency-Accuracy Tradeoff** We quantify the efficiency of each method based on the **number of summarizer calls** made during proposer selection (i.e., the sample complexity). All baselines, as well as Truth-Prediction Greedy, rely on proposers' individual reported labels and thus incur zero summarizer calls. Oracle-Surrogate Greedy approximates the summarizer's accuracy using a training set and requires $(k+1)T_{\hat{f}}$ calls, where $k+1$ indexes the possible counts of correct inputs and $T_{\hat{g}}$ denotes the number of Monte

Table 3: Sample complexity of considered methods, measured as the number of summarizer queries during proposer-selection.

| Method | Complexity |
|---|---|
| All baselines | 0 |
| Truth-Prediction Greedy | 0 |
| Oracle-Surrogate Greedy | $(k+1)T_{\hat{g}}$ |
| Model-First Greedy | $n_{\text{LLM}}\, m\, k$ |

Carlo samples per case; in our experiments $T_{\hat{g}} = 200$, this leads to 1,200 calls in total. Model-First Greedy queries calling the summarizer in each round to iterate all models, which scales as $n_{\text{LLM}} \cdot m \cdot k$; in our experiments, this leads to $8 \cdot 400 \cdot 5 = 16{,}000$ calls.

## 4.3 PROMPTING SUMMARIZER

In this subsection, we evaluate how (i) the order of proposer inputs and (ii) whether we include their individual accuracies influence the summarizer accuracy.

We pick five proposers with relatively large accuracy differences, and input their responses to a summarizer based on ascending, descending, and randomized order in their individual accuracies. For each case, we further distinguish two settings depending on whether the accuracy of each proposer is input to the summarizer as a part of the prompt.

Table 4 presents an example with two key takeaways. First, **inputting accuracy matters.** Providing per-proposer accuracies affects performance in opposite ways across datasets—improving on AIME yet degrading on CLadder. This suggests that the LLM summarizer can respond to the "reliability" information, but the net effect is heavily context-dependent. Second, **ordering matters.** Placing stronger proposers later in the prompt—i.e., using ascending accuracy order—outperforms descending order. This pattern is consistent with recency bias in long-context LLM inference (Peysakhovich & Lerer, 2023): earlier content tends to receive less attention relative to later content. These findings help clarify why our main experiments adopted a randomized ordering with per-proposer accuracies.

Table 4: Summarizer accuracies under different proposer orderings and whether individual accuracies are input in the (AIME or CLadder, ·, Ace, $k = 5$) setting with proposers: QwQ, Gemini, Llama, GPT, Aya, under instruction prompt 1 (Appendix C).

| Ordering | AIME | | CLadder | |
|---|---|---|---|---|
| | Without accuracy | With accuracy | Without accuracy | With accuracy |
| Ascending | 0.524 | 0.526 | 0.806 | 0.773 |
| Descending | 0.500 | 0.496 | 0.792 | 0.759 |
| Randomized | 0.498 | 0.538 | 0.798 | 0.774 |

## 5 CONCLUSION AND DISCUSSION

In this paper, we propose **complementary-MoA**—a proposer-selection framework for post-inference LLM ensembles that explicitly optimizes team effects between proposers and the summarizer. Focusing on multiple-choice QA, we connect the problem to feature selection over a black-box objective and instantiate three algorithms that realize different accuracy–efficiency trade-offs. Experiments confirm the robustness of our methods, clarify when and why accuracy- or diversity-based baselines fail, and suggest prompting strategies that further strengthen multi-AI collaboration. Nonetheless, we acknowledge several limitations and future directions. First, our label-level complementary algorithms work under multiple-choice scenarios, while the framework extends to open-ended tasks whenever a reliable evaluation metric is available. Second, the efficiency–accuracy frontier is not yet fully charted—hybrid designs (e.g., carefully choosing the first $k' < k$ proposers, then filling the remainder by accuracy) may further reduce sample complexity.

## ETHICS STATEMENT

Our study evaluates post-inference LLM ensembling on public, non-personal benchmarks and involves no human subjects or sensitive data; IRB approval was not required. Potential risks of the ensemble framework include amplifying biases present in base proposer models and misuse of ensembles; we mitigate these by explaining the mechanisms behind various methods, avoiding sensitive deployment claims, and providing new methods with significant improvements in robustness. The authors report no conflicts of interest or sponsorship that could inappropriately influence this work.

## REPRODUCIBILITY STATEMENT

We commit to enabling independent verification and the reproducibility of our experiments. In the paper, we specify the objective and detailed procedures for every proposed method, and all LLM usage, datasets, and experiment setups are included. In terms of the computation environment, all experiments run on non-proprietary models used 4x Nvidia RTX 4500 Ada GPUs or 1x Nvidia RTX 4500 Ada GPU and 3x Nvidia A100 Ampere.

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

## A    ADDITIONAL RELATED WORK

**Agent-Level Ensemble.**    LLM ensembles can be constructed at multiple stages of the inference pipeline (Chen et al., 2025b). We focus on *agent-level* ensembling, which treats each LLM as a black box. A closely related paradigm is *mixture-of-agents (MoA)* (Wang et al., 2024), a layered collaboration scheme in which, at a given layer, multiple proposers submit responses that are then aggregated by a summarizer. Wang et al. (2024) show that MoA effectively aggregates complementary signals, often yielding more reliable outputs than a single stronger model. A follow-up study challenges this design by demonstrating that repeatedly querying a single powerful LLM can also boost MoA-style performance (Li et al., 2025). Another line of related work is *LLM debate* (Du et al., 2023; Estornell & Liu, 2024; Chan et al., 2023; Wang et al., 2023a), where multiple models iteratively critique and refine one another's that can often result in a consensus outperforming a single model. However, as Estornell & Liu (2024) point out, sharing all agents' responses is not always optimal, where they observe that select a subset of LLMs that maximizes the mutual information between agents can be more effective. Our work targets the foundational step in the above frameworks—the $N \to 1$ summarization—with particular emphasis on proposing a more principled way to decide which proposers to select for the summarizer.

**Training-Based Ensemble.**    Prior literature has also explored the idea of training parametric meta-models to decide, per query, which LLM (or which LLM's output) to trust. For example, fusion methods train a small network on features from multiple LLMs—e.g., concatenated probabilities or last-layer embeddings—to predict the true label (Jiang et al., 2023; Wang et al., 2023b). Routing methods learn a delegator that selects the most suitable agents for various tasks, e.g., RouteLLM uses human preference data to bette trade off cost and quality (Lu et al., 2023), and ZOOTER learns a router based on distilling rewards on training queries Lu et al. (2023). Similarly, cost-aware cascades like FrugalGPT focus on learning when to use stronger but more expensive models (Chen et al., 2023). Unlike prior training-based ensembles, our framework avoids substantial supervised datasets: a few hundred examples suffice to learn the summarizer's behavior for better proposer selection. This light training also makes it compatible with closed-source LLM summarizers, whereas past work either does not use an LLM summarizer or requires open-source access (e.g., logits/weights).

**Feature-Selection.**    Our problem is naturally relevant to feature selection, where the goal is to select a small subset of features that optimize the performance of an ML model. One of the most classic example is Wrapper (Kohavi & John, 1998), which evaluates features by repeatedly training a model—using forward/backward search. The selection of features can also be implemented by inducing sparsity during training, with examples like LASSO (Tibshirani, 1996) and LARS (Kolter & Ng, 2009). Furthermore, it is often beneficial to filter likely weak features without retraining a predictor based on information-theoretic (e.g., mRMR (Peng et al., 2005)) or neighborhood criteria (e.g., Relief (Urbanowicz et al., 2018)). However, two challenges limit applicability to our setting. First, wrapper-style methods demand extensive retraining, while filter/embedded approaches operate only at the label level and thus ignore the LLM summarizer's error-correcting behavior.[3] Second, the summarizer's performance is often non-monotone in the set of agents, making standard marginal-gain scoring unreliable; this motivates new evaluation metrics for agent contribution—e.g., our $k$-greedy algorithm in Section 3.2.

---

[3]That said, in Section 4.2 we show that even label-level selection can produce more robust ensembles than existing baselines.

# B ADDITIONAL RESULTS

## B.1 LABEL-LEVEL AGGREGATION

In Table 5 and 6, we first present the accuracy of each proposer while answering the questions independently. As we can see, QwQ is an outstanding model in comparison to others; Aya is a weak model as a proposer, while we observe that it is a fast and accurate summarizer.

Table 5: Independent accuracy for each proposer with rows indicating models, columns indicating prompt IDs on the AIME dataset.

| Model | 1 | 2 | 3 | 4 | 5 |
|---|---|---|---|---|---|
| GPT-4o | 0.424 | 0.418 | 0.366 | 0.390 | 0.436 |
| AceReason-Nemotron-14B | 0.444 | 0.482 | 0.468 | 0.452 | 0.442 |
| Llama-3.3-70B-Instruct | 0.484 | 0.476 | 0.460 | 0.466 | 0.450 |
| QwQ-32B | 0.500 | 0.502 | 0.480 | 0.536 | 0.578 |
| Qwen3-32B | 0.452 | 0.506 | 0.460 | 0.468 | 0.494 |
| Sky-T1-32B-Preview | 0.408 | 0.430 | 0.420 | 0.410 | 0.416 |
| aya-expanse-32b | 0.236 | 0.264 | 0.252 | 0.242 | 0.266 |
| Gemini1.5-pro | 0.488 | 0.480 | 0.490 | 0.496 | 0.490 |

Table 6: Independent accuracy for each proposer with rows indicating models, columns indicating prompt IDs on the CLadder dataset.

| Model | 1 | 2 | 3 | 4 | 5 |
|---|---|---|---|---|---|
| GPT-4o | 0.681 | 0.680 | 0.682 | 0.677 | 0.685 |
| AceReason-Nemotron-14B | 0.708 | 0.726 | 0.692 | 0.707 | 0.726 |
| Llama-3.3-70B-Instruct | 0.507 | 0.595 | 0.502 | 0.538 | 0.532 |
| QwQ-32B | 0.775 | 0.789 | 0.803 | 0.802 | 0.797 |
| Qwen3-32B | 0.678 | 0.717 | 0.710 | 0.707 | 0.742 |
| Sky-T1-32B-Preview | 0.599 | 0.587 | 0.591 | 0.583 | 0.575 |
| aya-expanse-32b | 0.526 | 0.548 | 0.527 | 0.511 | 0.519 |
| Gemini1.5-pro | 0.707 | 0.704 | 0.718 | 0.734 | 0.748 |

We further test label-level aggregators against LLM summarizer aggregation, aiming to show that leveraging proposers' textual reasoning can boost accuracy. We evaluate three majority-vote variants: (i) over all proposers, (ii) over the best prompt per model, and (iii) over the best model per prompt. We also include weighted majority vote, using the classic log-odds weights $w_i \propto \log \frac{p_i}{1-p_i}$ derived for independent binary voters by Nitzan & Paroush (1982). Although our setting involves multiclass labels and correlated voters, we adopt this weighting as a heuristic baseline. Finally, we consider a learning-based baseline that trains a decision tree on the $N$ proposers' labels (features) to predict the ground truth, and report test accuracy.

As shown in Tables 7 and 8, majority-vote baselines perform competitively with the LLM summarizers on the binary CLadder dataset (a setting that prefers simple majority vote), yet they are substantially outperformed by LLM summarizers on the multichoice AIME dataset. The decision-tree baseline is dominated by majority voting on both datasets. These results underscore the importance of incorporating textual evidence from proposers' reasoning, rather than relying solely on label-level aggregation.

## B.2 COMPARING PROPOSER SELECTION METHODS (CONTINUED)

Here, we present the comparison of methods in other settings, aiming to show the robustness of our methods. To be consistent, we present all results in the same format as Table 1 and 2.

Table 7: Label-level aggregation baselines on CLadder.

| Method | Accuracy | |
| --- | --- | --- |
| | Unweighted | Weighted |
| Majority | 0.795 | 0.814 |
| Majority (best prompt per model) | 0.816 | 0.816 |
| Majority (best model per prompt) | 0.793 | 0.805 |
| Decision Tree | 0.737 | |

Table 8: Label-level aggregation baselines on AIME.

| Method | Accuracy | |
| --- | --- | --- |
| | Unweighted | Weighted |
| Majority | 0.726 | 0.734 |
| Majority (best prompt per model) | 0.632 | 0.628 |
| Majority (best model per prompt) | 0.67 | 0.676 |
| Decision Tree | 0.612 | |

**Ensemble Size** Table 10 and 9 report results for the (AIME, with QwQ, Aya) setting at $k \in \{3, 4\}$ settings. Note that the results for Input-all, Best-model, and MoA remain the same, as their performance does not depend on $k$. Our results confirm the robustness of our complementary-MoA framework, as it remains competitive with the strongest baselines; the only notable exception is that *Top-accuracy* is unusually strong at $k = 3$. Overall, we do not observe a monotonic improvement in summarizer accuracy as the ensemble size increases.

Table 9: A comparison of methods in the (AIME, with QwQ, Aya, $k = 3$) setting.

| Method | Average counts of selected proposers | | | | | | | | Accuracy |
| --- | --- | --- | --- | --- | --- | --- | --- | --- | --- |
| | QwQ | Qwen | Llama | Gemini | GPT | Sky | Aya | Ace | |
| Input-all | 5 | 5 | 5 | 5 | 5 | 5 | 5 | 5 | 0.746 |
| Best-model | 5 | — | — | — | — | — | — | — | 0.766 |
| Top-accuracy | 2 | 1 | — | — | — | — | — | — | **0.792** |
| MoA | 1 | 1 | 1 | 1 | 1 | 1 | 1 | 1 | 70.0 |
| Conditioned-diversity | 1 | — | — | — | — | 1 | 1 | — | 0.519 |
| **Truth-prediction Greedy** | 1 | 2 | — | — | — | — | — | — | 0.714 |
| **Oracle-surrogate Greedy** | 1 | — | 1 | — | 1 | — | — | — | 0.728 |
| **Model-first Greedy** | 2 | 1 | — | — | — | — | — | — | 0.744 |

**Summarizer and Dataset** Here, we present the analogous results on the binary QA dataset CLadder, with Ace being the summarizer. In addition to proving the robustness of our methods, we highlight the following observations. First, as we have seen in the main body, Input-all has been a baseline that works well with Aya being the summarizer. However, our results with AceReason being the summarizer challenge its robustness (see Table 11). We further emphasize that Input-all requires significantly long-context inputs, and thus requires much longer inference time.

Second, as discussed in the main text, all methods perform similarly on the Cladder dataset, so seeking complementary proposer teams yields only modest gains. Two factors likely explain this: (i) the binary label space leaves little room for error correction, and (ii) roughly 20% of questions are difficult and are missed by most of the proposers, which caps the potential benefit of aggregation.

Table 10: A comparison of methods in the (AIME, with QwQ, Aya, $k = 4$) setting.

| Method | Average counts of selected proposers | | | | | | | | Accuracy |
| --- | --- | --- | --- | --- | --- | --- | --- | --- | --- |
| | QwQ | Qwen | Llama | Gemini | GPT | Sky | Aya | Ace | |
| Input-all | 5 | 5 | 5 | 5 | 5 | 5 | 5 | 5 | 0.746 |
| Best-model | 5 | — | — | — | — | — | — | — | 0.766 |
| Top-accuracy | 3 | 1 | — | — | — | — | — | — | 0.740 |
| MoA | 1 | 1 | 1 | 1 | 1 | 1 | 1 | 1 | 0.660 |
| Conditioned-diversity | 1 | — | 1 | — | 1 | 1 | — | — | 0.620 |
| **Truth-prediction Greedy** | 2 | 2 | — | — | — | — | — | — | 0.740 |
| **Oracle-surrogate Greedy** | 2 | 1 | — | 1 | — | — | — | — | 0.728 |
| **Model-first Greedy** | 1 | 1 | — | — | — | — | — | 2 | **0.796** |

Table 11: A comparison of methods in the (AIME, with QwQ, **Ace** summarizer, $k = 5$) setting.

| Method | Average counts of selected proposers | | | | | | | | Accuracy |
| --- | --- | --- | --- | --- | --- | --- | --- | --- | --- |
| | QwQ | Qwen | Llama | Gemini | GPT | Sky | Aya | Ace | |
| Input-all | 5 | 5 | 5 | 5 | 5 | 5 | 5 | 5 | 0.339 |
| Best-model | 5 | — | — | — | — | — | — | — | 0.540 |
| Top-accuracy | 3 | 1 | — | 1 | — | — | — | — | 0.559 |
| MoA | 1 | 1 | 1 | 1 | 1 | 1 | 1 | 1 | 0.481 |
| Conditioned-diversity | 1 | — | 1 | — | — | 1 | 2 | — | 0.507 |
| **Truth-prediction Greedy** | 1 | 3 | — | — | — | 1 | — | — | 0.542 |
| **Oracle-surrogate Greedy** | 2 | 1 | 1 | — | — | 1 | — | — | **0.561** |
| **Model-first Greedy** | — | — | 2 | — | — | — | — | 3 | 0.553 |

Table 12: A comparison of methods in the (AIME, **without** QwQ, **Ace** summarizer, $k = 5$) setting.

| Method | Average counts of selected proposers | | | | | | | Accuracy |
| --- | --- | --- | --- | --- | --- | --- | --- | --- |
| | Qwen | Llama | Gemini | GPT | Sky | Aya | Ace | |
| Input-all | 5 | 5 | 5 | 5 | 5 | 5 | 5 | 0.399 |
| Best-model | — | — | 5 | — | — | — | — | 0.564 |
| Top-accuracy | 1 | 1 | 2 | — | — | — | — | 0.508 |
| MoA | 1 | 1 | 1 | 1 | 1 | 1 | 1 | 0.452 |
| Conditioned-diversity | 1 | 1 | — | — | 1 | 2 | — | 0.522 |
| **Truth-prediction Greedy** | 3 | — | 1 | 1 | — | — | — | 0.534 |
| **Oracle-surrogate Greedy** | 1 | 1 | 2 | — | — | — | 1 | 0.564 |
| **Model-first Greedy** | — | 1 | 2 | — | — | — | 2 | **0.594** |

**Distribution of Correct Answers** Figure 3 presents the analogous illustrative example for the remaining methods.

### B.3 PROMPTING SUMMARIZERS

Here, we present the results analogous to Table 4 with Aya being the summarizer. As we can observe, the same patterns hold.

Table 13: A comparison of methods in the (Cladder, with QwQ, Ace, $k = 5$) setting.

| Method | Average counts of selected proposers | | | | | | | | Accuracy |
| | QwQ | Qwen | Llama | Gemini | GPT | Sky | Aya | Ace | |
| --- | --- | --- | --- | --- | --- | --- | --- | --- | --- |
| Input-all | 5 | 5 | 5 | 5 | 5 | 5 | 5 | 5 | 0.804 |
| Best-model | 5 | — | — | — | — | — | — | — | 0.736 |
| Top-accuracy | 3 | — | — | — | — | — | — | — | 0.739 |
| MoA | 1 | 1 | 1 | 1 | 1 | 1 | 1 | 1 | 0.766 |
| Conditioned-diversity | 1 | — | — | — | — | — | 2 | — | 0.742 |
| **Truth-prediction Greedy** | 3 | — | — | — | — | — | — | — | 0.734 |
| **Oracle-surrogate Greedy** | 3 | — | — | — | — | — | — | — | 0.742 |
| **Model-first Greedy** | — | — | — | 3 | 2 | — | — | — | **0.811** |

Table 14: A comparison of methods in the (Cladder, without QwQ, Ace, $k = 5$) setting.

| Method | Average counts of selected proposers | | | | | | | Accuracy |
| | Qwen | Llama | Gemini | GPT | Sky | Aya | Ace | |
| --- | --- | --- | --- | --- | --- | --- | --- | --- |
| Input-all | 5 | 5 | 5 | 5 | 5 | 5 | 5 | 0.792 |
| Best-model | — | — | 5 | — | — | — | — | **0.796** |
| Top-accuracy | 1 | — | 2 | — | — | — | 2 | 0.718 |
| MoA | 1 | 1 | 1 | 1 | 1 | 1 | 1 | 0.772 |
| Conditioned-diversity | — | 1 | 1 | — | — | — | 1 | 0.760 |
| **Truth-prediction Greedy** | — | 1 | 1 | — | 1 | — | — | 0.756 |
| **Oracle-surrogate Greedy** | 1 | — | 1 | — | — | — | 1 | 0.763 |
| **Model-first Greedy** | — | — | 3 | 2 | — | — | — | 0.792 |

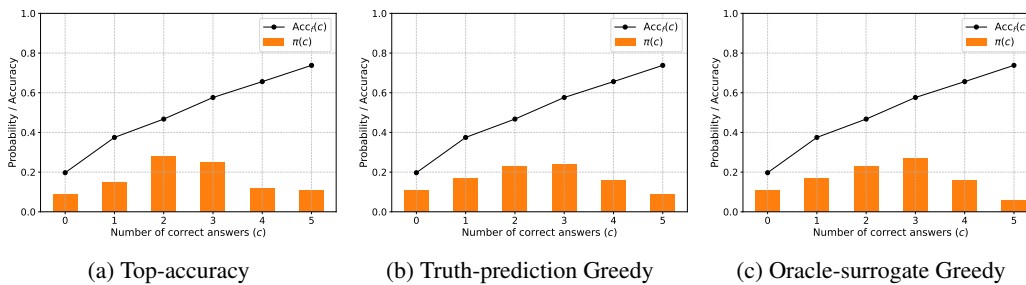

(a) Top-accuracy  (b) Truth-prediction Greedy  (c) Oracle-surrogate Greedy

Figure 3: Distribution of the number of correct answers (bars) and summarizer accuracy $\mathrm{Acc}_f(c)$ (line) for the remaining methods, complementing Fig. 2, in the (AIME, without QwQ, Aya, $k = 5$) setting.

Table 15: Summarizer accuracies under different proposer orderings and whether individual accuracies are input in the (AIME or CLadder, ·, Aya, $k = 5$) setting under instruction prompt 1 (Appendix C).

| | AIME | | CLadder | |
| Ordering of proposers | Without accuracy | With accuracy | Without accuracy | With accuracy |
| --- | --- | --- | --- | --- |
| Ascending | 0.574 | 0.618 | 0.693 | 0.605 |
| Descending | 0.578 | 0.596 | 0.658 | 0.591 |
| Randomized | 0.526 | 0.568 | 0.683 | 0.587 |

## C  PROMPTS

---

**Multi-choice — Proposer Prompt**

You will solve a multiple choice question. Format your answer to include:
1. A full response
2. A concise step-by-step reasoning
3. The single letter choice

---

**Binary-choice — Proposer Prompt**

You will answer a yes or no question. Format your answer to include:
1. A full response
2. A concise step-by-step reasoning
3. The yes or no answer

---

**Multi-choice — Summarizer Prompt**

I will give you a multiple choice question and potential solutions that may be correct or incorrect. Your task is to analyze the reasoning of the potential solutions step by step.
If there are any errors, correct them and update your answer.
If there are no errors, answer the question matching those solutions.
Your answer must be in the format of a full response, then a letter choice.

---

**Binary-choice — Summarizer Prompt**

I will give you a yes or no question and multiple potential solutions that may be correct or incorrect. Your task is to analyze the reasoning of the potential solutions step by step.
If there are any errors, correct them and update your answer.
If there are no errors, answer the question matching those solutions.
Your answer must be in the format of a full response, then a yes or no answer.

---

**Instruction Prompt 1**

Divide the question into smaller, manageable parts and tackle each part individually before synthesizing the overall answer.

---

**Instruction Prompt 2**

Use mathematical principles and logic to solve the problem, even if it's not a math question.

---

**Instruction Prompt 3**

Relate the question to a familiar concept or situation to better understand and solve it.

---

**Instruction Prompt 4**

Think about what the answer would be if the opposite were true, to gain a different perspective.

---

**Instruction Prompt 5**

Eliminate the obviously incorrect answers first and then choose the most likely correct answer.

## D   LLM USAGE

Large language models (LLMs) were used in this paper only as a general-purpose writing assistant. Specifically, they supported adjusting phrasing for clarity, polishing grammar, shortening sentences, and reformatting text. LLMs were also used to generate and refine tables (e.g., aligning multi-column headers and converting between LaTeX table styles). At no point did LLMs contribute to research ideas, conceptual framing, or experimental design. All substantive intellectual contributions are solely those of the authors.

