# OpenReview forum: "Mixture of Complementary Agents for Robust LLM Ensemble"
_ICLR.cc/2026/Conference — ICLR 2026 Conference Withdrawn Submission_

### Official Review · Reviewer_TV88 · 2025-10-25

**Soundness:** 2
**Presentation:** 2
**Contribution:** 2
**Rating:** 4
**Confidence:** 5

**Summary:**

The paper studies how to select a subset of proposer models for an LLM-based summarizer/aggregator, defining the objective around “complementarity” rather than individual accuracy or surface diversity. It instantiates this with three greedy procedures: Model-first Greedy, Truth-prediction Greedy, and Oracle-surrogate Greedy. Experiments on a multiple-choice reformulation of AIME and the CLadder binary classification set assess performance across different summarizers, candidate pools, subset sizes, and prompt-order/metadata settings.

**Strengths:**

1.	Clear and accessible writing.
2.	The paper formulates the task of choosing a subset of proposer models for an LLM summarizer around the idea of complementarity and provides concrete greedy procedures that are straightforward to implement.

**Weaknesses:**

1.	The core idea of optimizing for complementarity rather than raw accuracy or diversity has already been explored in agent-level ensemble frameworks [A, B]. The paper does not clearly articulate what is substantively new relative to these lines of work, nor does it position its contributions with sufficient precision.
2.	The evaluation is limited in dataset coverage. To substantiate generality, the study should include more diverse benchmarks, for example, GPQA for difficult factual reasoning and LiveCodeBench for program synthesis. The current evidence is insufficient to claim broad applicability.
3.	The decoding configuration undermines reproducibility. The default temperature is set to 0.7, which introduces randomness and makes exact replication difficult. The paper should either set the temperature to 0 for deterministic decoding or provide a rigorous protocol that fixes all sampling parameters and random seeds, with variance reported across multiple runs.
4.	The set of baselines is incomplete. Recent LLM ensemble methods have become increasingly numerous, and comparisons should include stronger methods such as LLM-Blender [C] and Reconcile [D].
5.	Typo: “submodelar” → “submodular”.
6.	Table anomalies/inconsistencies (e.g., a “70.0” entry for MoA accuracy in Table 9; inconsistent decimal precision in Table 8) need cleanup.

[A] Efficient Dynamic Ensembling for Multiple LLM Experts, IJCAI 2025.

[B] Fusing models with complementary expertise, ICLR 2024.

[C] Llm-blender: Ensembling large language models with pairwise ranking and generative fusion, ACL 2023.

[D] ReConcile: Round-Table Conference Improves Reasoning via Consensus among Diverse LLMs, ACL 2024.

**Questions:**

see Weaknesses

---

### Official Review · Reviewer_vkLm · 2025-10-27

**Soundness:** 2
**Presentation:** 2
**Contribution:** 2
**Rating:** 4
**Confidence:** 4

**Summary:**

The paper is about an ensemble approach of LLMs.\
Specifically, we have multiple proposers and a summarizer.\
Our goal is to select the best combination (subset) of proposers for the summarizer.\
To this end, the authors find the best (global) subset on the validation set, then use that for the inference.

**Strengths:**

1. The motivation is acceptable, and an important problem in my opinion.
2. The authors conduct an experiment on public datasets to demonstrate the superiority of the proposed method.

**Weaknesses:**

1. Title: MIXTURE OF COMPLEMENTARY AGENTS FOR ROBUSTLLM ENSEMBLE
- Is it really for the LLM agents?
- I think the paper is about just vanilla LLMs.

2. Method
- Greedy algorithm: how to measure the accuracy for test inputs? We do not have labels for them.\
If you use m labeled validation set for selecting S from N proposers, then we have a "global" set of proposers for all text inputs.\
However, I think each test input needs a different combination of proposers.
- MODEL-FIRST GREEDY is just a simple random sampling to reduce summarizer calls per iteration.\
I think it lacks technical contribution.
- Oracle-Surrogate Greedy: do we really need a ML model for g : {0, . . . , k} → [0, 1]?\
I think we can just count the number of accurate final predictions, for each number of correct proposers.

3. Experiment
- Comparison with other test-time scaling approaches would be beneficial.
- In Table 1,2, there are no decimal points. Other models are not selected for a single time among 500/1k questions?
- "Oracle" method would be beneficial to be included in the comparison, just for the reference of an upper bound of this approach.

**Questions:**

Please refer to Weaknesses.

---

### Official Review · Reviewer_kuzG · 2025-10-30

**Soundness:** 2
**Presentation:** 2
**Contribution:** 2
**Rating:** 4
**Confidence:** 3

**Summary:**

- Introduces **complementary-MoA**, a framework for proposer selection in LLM ensemble settings. Rather than prioritising individual accuracy or diversity, it aims to select proposers that *complement* each other and the summariser.
- Three algorithms are proposed with different accuracy–efficiency trade-offs:
  1. **Model-first greedy:** selects proposers iteratively based on summariser-validated marginal accuracy.
  2. **Truth-prediction greedy:** trains a small model on proposers’ labels to estimate complementarity.
  3. **Oracle-surrogate greedy:** approximates summariser behaviour based on how many correct labels each subset provides.
- Evaluated mainly on **AIME** (multi-choice math) and **CLadder** (binary causal reasoning) using 8 LLMs (Qwen3, Llama-3.3, GPT-4o, Gemini, Aya, etc.). Results show modest improvements, mostly when the top model (QwQ) is included.

**Strengths:**

- **Principled framing:** Explicitly formulates proposer selection as an optimisation over complementarity rather than heuristic accuracy/diversity balancing.
- **Algorithmic variety:** Provides multiple variants addressing sample complexity trade-offs; model-first greedy yields noticeable empirical gain.
- **Clear presentation:** Methods are systematically described, and ablations (e.g. with/without QwQ, varying k) are well structured.

**Weaknesses:**

- **Limited contribution:** The main conceptual step (proposer complementarity) feels incremental to existing ensemble selection ideas; the practical novelty beyond “accuracy+diversity trade-off” is thin.
- **Questionable baseline accuracy:** Figure 1 reports only **44.4%** on AIME for **Qwen3-32B**, yet the same model is publicly documented at **≈70%** on AIME 2025. This discrepancy raises concerns about dataset setup, prompt formatting, or evaluation procedure. Without clarification, the comparative gains are difficult to interpret.
- **Weak empirical scope:** Only two datasets (AIME, CLadder) are tested; no qualitative examples show why “complementary” proposers improve reasoning quality.
- **Dependence on validation set:** The approach requires a small labelled calibration set to fit combination weights. Generalisation to new proposer sets or tasks is untested.
- **Lack of strong baselines:** Comparisons omit stronger ensemble or debate-based baselines (e.g., LLM-Blender, MoA with reranking, or single high-capacity models with multiple sampling).
- **Scalability & cost:** Methods that keep summarisers in the loop (e.g. model-first greedy) are computationally heavy; efficiency analysis is limited to relative sample counts rather than runtime or GPU cost.

**Questions:**

1. **Baseline discrepancy:** Figure 1 reports 44.4% for Qwen3-32B on AIME, yet recent public results show ≈70%. Could you clarify if this is due to dataset versioning, different prompt format, or restricted evaluation (e.g., no CoT)?
2. **Generalisation:** The algorithms require a labelled validation set to tune complementarity. How robust are these learned proposer combinations to new model pools or unseen tasks?
3. **Scope:** Beyond AIME and CLadder, have you tried reasoning or factuality datasets (e.g., GSM8K, ARC-C, MMLU)?
4. **Qualitative analysis:** Can you provide examples showing that selected proposers add genuinely complementary reasoning or knowledge rather than random variance?
5. **Fairness of baselines:** Since complementary-MoA jointly fine-tunes and infers with multiple LLMs, have you compared against a *single* larger model with similar total parameter count (≈ n × base size)?
6. **Complexity:** What are the end-to-end runtime and memory costs for model-first greedy compared with standard MoA or self-MoA ensembles?

---

### Official Review · Reviewer_LYcB · 2025-11-01

**Soundness:** 2
**Presentation:** 2
**Contribution:** 2
**Rating:** 2
**Confidence:** 4

**Summary:**

This paper introduces a framework for ensembling LLM proposer agents based on their complementarity with the summarizer agent, rather than just individual accuracy or output diversity. The authors proposes several greedy algorithms with different cost-accuracy trade-offs for proposer selection, including:
* _Model-First Greedy_, which is a simplified version of the standard greedy algorithm that estimate proposer's marginal accuracy at model level;
* _Truth-Prediction Greedy_, which trains a lightweight predictor $g_\theta$ on a labeled validation set to predict the ground-truth label from a set of proposers’ output labels, and use this predictor to estimate the summarizer's performance;
* _Oracle-Surrogate Greedy_, which uses a simplified surrogate model $\tilde{g}(c)$, depending only on the number of correct inputs $c$ out of $k$ proposers, to approximate summarizer performance in a Monte Carlo manner.

Experiments on multi-choice AIME for math reasoning and CLadder for binary causal reasoning show that complementarity-based selection methods perform better and have more robust performance against baselines like choosing top-accuracy models or maximizing diversity. The paper also analyzes how proposer ordering and exposing per‑proposer accuracy to the summarizer affect results.

**Strengths:**

1. The complementarity framing is well-motived with an illustrative to show the performance gap between most accurate and most complementary proposer for the same summarizer.
2. The proposed complementarity‑based selection methods perform robustly across diverse settings—both with and without a dominant (“dictator”) proposer, across summarizers of varying capability, and on benchmarks of differing difficulty.

**Weaknesses:**

1. **Vague algorithm specification**. The _Truth-Prediction Greedy_ algorithm is only described in high-level terms, lacking crucial implementation details. The authors state that $g_\theta$ is a lightweight ML model that is trained to predict the true answer (i.e., $Y_q$). While I can understand that $g_\theta$ is used as an approximation of the summarizer agent. But what type of ML model for $g_\theta$ is being used in the experiment and how it is exactly being trained? There is also no discussion of how sensitive the method is to the capacity or mis-specification of $g_\theta$ (e.g. if $g_\theta$ underfits or the trained $g_\theta$ actually performs very different compared to the summarizer agent). These omissions make it difficult to reproduce the results and fully understand the conditions under which _Truth-Prediction Greedy_ works. Overall, the algorithm is presented in a somewhat abstract manner, undermining clarity and reproducibility.
2. **Questionable surrogate model assumption**. The surrogate function $\tilde{g}(c)$ in the _Oracle-Surrogate Greedy_ algorithm is confusing. It estimates the summarizer's success only by the number of correct inputs $c$ out of $k$, ignoring which proposers are being used. By not distinguishing proposer identity, $\tilde{g}(c)$ imposes a strong simplifying assumption that may not hold. The paper neither justifies this assumption theoretically nor tests its impact empirically (e.g., by checking whether summarizer accuracy varies significantly with different combinations yielding the same $c$).
3. **Using of per-proposer accuracy in prompt**. In the experiments, the authors feed the summarizer a list of proposers’ answers along with statements of how often each proposer is correct (based on a held-out set for proposer selection), which is essentially giving the summarizer strong prior knowledge about which sources are more trustworthy. This is unrealistic for real‑world scenarios where ground‑truth accuracies for each agent cannot typically be obtained ahead of time. An ablation that removes per‑proposer accuracy from the prompt across all experimental settings would be important but is missing.
4. **Multi-choice AIME**. The AIME dataset used for evaluation is artificially constructed for multi-choice QA in a way that may inflate performance as the wrong choices are randomly generated. It is unclear why experiments are restricted to multiple‑choice QA when open‑ended math QA also supports accuracy measurement. I recommend running experiments on the original (open‑ended) AIME data and including established multiple‑choice QA benchmarks (e.g., MMLU), as used by the self‑MoA paper [1].
5. **Clarity and reproducibility of writing**. As noted above, Section 3.2 (greedy algorithms for complementarity-based proposer selection) and the presentation of Algorithm 1 are difficult to follow. Definitions of different set functions (i.e., $\hat{Acc}$ and its variants) are highly compressed without enough textual explanations. There are also minor typos and table misalignments. For example, in Table 13 and Table 14, $k$ is set to be 5, but the sum of counts of selected proposers for _Conditioned-diversity_, _Truth-prediction Greedy_, and _Oracle-surrogate Greedy_ are all 3; in Table 9, MoA's accuracy should be written as 0.700 rather than 70.0.

[1] Li, Wenzhe, et al. "Rethinking Mixture-of-Agents: Is Mixing Different Large Language Models Beneficial?." arXiv preprint arXiv:2502.00674 (2025).

**Questions:**

Please see the weakness section. In addition, I have the following questions:

1. Seems that the final accuracy highly depends on the summarizer. For example, according to Table 5 and Table 6, Ace is significant stronger LLM compared to Aya. However, using Ace as the summarizer (Table 11 and Table 12) demonstrates significantly worse performance compares to using Aya as the summarizer (Table 1 and Table 2). Why is this happening? Besides, as the final accuracy highly depends on the selection of the summarizer, it would be beneficial to conduct the experiments on other summarizer models to demonstrate the effectiveness and robustness of the proposed complementarity-based methods.
2. Since each configuration averages 10 summarizer calls per question, please report standard deviations and/or 95% confidence intervals, and perform significance tests for the differences in accuracy between methods, to verify that the observed gains are statistically robust.

---

### Official Review · Reviewer_G9RU · 2025-11-01

**Soundness:** 2
**Presentation:** 2
**Contribution:** 2
**Rating:** 2
**Confidence:** 4

**Summary:**

This paper presents an approach that ensembles multiple LLM agents, referred to as proposers, along with a summarizer that produces the final output based on the proposers’ responses. The method estimates each proposer’s contribution using scores very similar to Shapley values and aims to identify an optimal subset of $K$ proposers that perform best collectively.

**Strengths:**

* The authors propose three heuristics for choosing the subset $S$ of proposers.

* They analyze and aim to reduce the sample complexity, focusing on the number of API calls required to the summarizer.

* Their method demonstrates performance improvements over several naive baseline approaches.

**Weaknesses:**

* My major concern is the **novelty** of this work. The authors claim that prior literature only considers the *diversity* and *accuracy* of individual LLM agents when forming multi-agent systems. This is not entirely accurate. There exists prior and ongoing research on estimating and scoring agents in multi-agent settings using **Shapley values** [1][2][3]. In fact, the approach proposed as their first contribution closely resembles the definition of Shapley values, which has already been explored. In the Shapley framework, each agent (or team member) is assigned a score based on how much its inclusion increases the overall performance. There are also various methods to approximate Shapley values without evaluating all possible subsets, **yet none of these works are mentioned or compared against**. Moreover, the paper does not acknowledge that selecting agents based on their marginal accuracy is directly similar to Shapley value computation. As a result, I find the “Previous Ideas” subsection and the stated contributions not entirely accurate, which raises concerns about the overall novelty of the work.

*  My other major question concerns how these algorithms operate in a **dynamic setting**. In practical scenarios, we may introduce a new set of agents into an already functioning multi-agent system and want to integrate them into the existing team, or conversely, remove some of the previous agents. Similarly, the task itself may evolve over time, requiring the system to adapt. However, the paper’s formulation seems to address only a *static setting*, where team formation is treated as a one-time preprocessing step; once the subset $S$ of agents is selected, it remains fixed. This assumption is not very realistic, since in real-world applications, new models are introduced frequently, often with different or improved capabilities. It would be valuable to discuss how the proposed approach could adapt to such dynamic environments.

* The experiments appear to be limited to relatively **simple tasks**, such as multiple-choice QA and binary QA. However, multi-agent settings are often applied in more complex domains, such as code generation, where the final output is not merely a straightforward ensemble or a single summarization of several similar agents. In such cases, agents may need to collaborate, specialize, or build upon each other’s outputs, which is far more challenging. The current experimental setup limits the contribution of the paper to a narrow and simplified scenario, reducing its broader applicability.

* My other question concerns whether there is any **theoretical analysis** of the proposed greedy algorithms, specifically, how well they can *approximate* the optimal subset $S^*$. For instance, is there any formal discussion on the trade-off between the *number of calls* to the summarizer (i.e., the budget) and the resulting *approximation quality* of the selected subset *S* compared to the optimal solution (in terms of $Acc$)? This question closely relates to the *approximation analysis of Shapley value estimation*, where similar results and guarantees have been studied.

*References:*

* [1] A Dynamic LLM-Powered Agent Network for Task-Oriented Agent Collaboration
* [2] Shapley-Coop: Credit Assignment for Emergent Cooperation in Self-Interested LLM Agents
* [3] An Adversary-Resistant Multi-Agent LLM System via Credibility Scoring

**Questions:**

See Weaknesses.

---

### Note · Authors · 2025-11-21

**Comment:**

We appreciate the reviewers' comments on improving the paper. After consideration, we have decided to withdraw the paper.

**Withdrawal Confirmation:**

I have read and agree with the venue's withdrawal policy on behalf of myself and my co-authors.